# Valorisation of Waste Bread for the Production of Yeast Biomass by *Yarrowia lipolytica* Bioreactor Fermentation

**Erdem Carsanba [1,2], Bilal Agirman [1], Seraphim Papanikolaou [3], Patrick Fickers [4] and Huseyin Erten [1,*]**

[1]  Faculty of Engineering, Department of Food Engineering, Cukurova University, 01330 Adana, Türkiye; ecarsanba@ucp.pt or carsanba.erdem@gmail.com (E.C.); bagirman@cu.edu.tr (B.A.)
[2]  CBQF—Centro de Biotecnologia e Química Fina—Laboratório Associado, Escola Superior de Biotecnologia, Universidade Católica Portuguesa, Rua Diogo Botelho 1327, 4169-005 Porto, Portugal
[3]  Department of Food Science & Human Nutrition, Agricultural University of Athens, 11855 Athens, Greece; spapanik@aua.gr
[4]  Gembloux Agro-Bio Tech, Microbial Processes and Interactions (MiPI), University of Liège, 5030 Gembloux, Belgium; pfickers@uliege.be
*  Correspondence: herten@cu.edu.tr

**Abstract:** The increase in the wastage of bread, representing 12.5 million tons per year, causes ecological problems, such as the production of methane and $CO_2$, when that waste bread (WB) is improperly managed. To reduce this ecological footprint, a more sustainable system of WB management must be set up. Based on its chemical composition, WB has a high potential to be used as feedstock for microbial growth and conversion into value-added bio products. The microbial valorisation of WB is a novel biotechnological approach to upgrading a waste into a renewable feedstock for bio-based industry, thus favouring the circular economy concept. Based on this, the aim of this study was to test WB as a feedstock for biomass production by *Yarrowia lipolytica*, which can be considered as a promising supplement for animal and human dietary products. The enzymatic hydrolysis of WB was primarily optimized for large-scale production in a bioreactor. The biomass production of *Y. lipolytica* strain K57 on WB hydrolysate-based media in batch bioreactor culture was then investigated. As a result, a very high starch to glucose conversion yield of 97% was obtained throughout optimised hydrolysis. At the end of 47 h of batch culture, a biomass higher than 62 g/L, specific growth rate of 0.37 h$^{-1}$ and biomass yield of 0.45 g/g were achieved from a WB hydrolysate. Therefore, this study demonstrates that WB hydrolysate has a promising potential to be used as a feedstock for biomass production by *Y. lipolytica* strain K57 for food and animal diet applications.

**Keywords:** waste valorisation; circular economy; enzymatic hydrolysis; yeast fermentation; *Yarrowia lipolytica* biomass; citric acid; microbial lipid

## 1. Introduction

Bakery products, which comprise the highest percentage in the nutrition pyramid, are the most common food all over the world. Bread, as the main bakery food, is a prime source of carbohydrates, is a ready-to-eat food, and can be consumed with any combination of any type of diet. The annual production of bread worldwide was estimated as 125 million tons [1]. Moreover, the production of 120 million loaves per day was reported in Turkey [2]. A significant amount of the produced bread is not sold from retail and is returned to the manufacturers due to the staling mechanism driven by the loss of moisture and retrogradation of starch, or microbial spoilage by the growth of mould and yeasts [3]. The increment in the quantity of waste bread (WB) was calculated as 10% of annual production, namely 12.5 million tons of waste bread per year worldwide and 12 million loaves wasted daily in Turkey [1,2]. Although a small part of this amount is used

as animal feed, high percentages are disposed of as waste. WB is harmful to the environment, and causes ecological problems such as the generation of biogas, mainly methane (60%), and carbon dioxide (40%). Therefore, WB contributes to air pollution with $CO_2$ emissions [4]. The WB accumulation rate and percentage are increasing continuously. The processing or disposing of WB is a major global problem. The first solution for this should be a reduction in waste bread through controlling the production rate and supply chain, and then other alleviating options can be considered. The bio-valorisation of this waste substrate into industrially important bio-based products might help to alleviate the environmental and economic problems. WB, which is very nitrogenous with high starch (more than 70% dry matter) and protein (up to 14% dry matter) contents [5], can be used as feedstock after hydrolytic enzyme pre-treatment to release the nutrients necessary for microbial growth and/or the production of high-value-added products. WB is a rich source of fermentable sugars with other nutrients, making it as a potential substrate for biorefineries. The fermentable sugars from WB are pure since they are free of inhibitors. Moreover, the recovery of sugars from WB is easier and more convenient than that of other crude renewable sources which need harsh pre-treatment processes, such as lignocellulosic biomass [6]. The homogenous and consistent composition of WB provides an advantage over other feedstocks. Therefore, the use of WB as feedstock for biorefineries will provide several advantages, including financial and economic benefits, to industries generating WB. The low cost of WB will also deliver financial benefits to biorefinery plants through the decreased costs associated with the use of a waste resource as a microbial feedstock [7]. In previous studies, WB has been used for the production of several value-added products such as biofuels [2,8–10], biohydrogen [11,12], organic acids [1], biomass [13], enzymes [14,15] and chemicals for food industries [3,16–18].

*Yarrowia lipolytica* is one of the most widely studied oleaginous and non-conventional yeasts due to its essential properties such as its generally recognized as safe (GRAS) status, high productivity, wide range of substrate usage, high tolerance to metal ions and salt solutions, and ability to grow at low and high pH (from 4 to 8) and in a wide range of temperatures (from 18 °C to 32 °C) [19]. Currently, this yeast has been used in the production of organic acids [20], single-cell oil (SCO) and fatty acids [21,22], sweeteners [23], enzymes [24], biomass [25], carotenoids [26], and antioxidants [27], as well as single-cell proteins (SCP) [28]. In addition, the use of *Y. lipolytica* as fodder yeast had been recently authorised by the European Feed Manufacturers' Federation (catalog number 00 575-EN) [29]. Yeast biomass is rich in protein (SCP), peptides, essential amino acids (e.g., lysine and methionine), fatty acids (e.g., PUFAs), vitamins and trace elements (e.g., macro elements: sodium, calcium, and micro elements: zinc or selenium) [29,30]. As a high-nutrient product, yeast biomass can be considered a promising replacement for animal diet products [30]. Moreover, the use of nutritional yeast biomass as a supplement to human diet can serve the solution for the shortage of food in continuously growing human population.

The *Y. lipolytica* strains tested in small-scale bioreactors often fail to show similar performance at industrial scale. The difference in production performance from small scale to commercial scale has generally been recognized as one of the main reasons for the limited commercialization success of microbial bioprocesses so far [31]. In industrial-scale bioreactors, there are several limitations such as poor mixing, heat transfer, and gas–liquid mass transfer, which result in the fluctuation of parameters such as dissolved oxygen, temperature, pH, medium composition, etc. [32]. There are few studies on the optimization of bioprocesses in *Y. lipolytica* in large-scale fermentation. Erythritol production by *Y. lipolytica* was performed in a pilot plant with a working volume of 0.5 m³, where the production reached 180.3 kg/L with a productivity of 1.25 kg/m³·h and a yield of 0.533 kg/kg, which was comparable with that at laboratory scale with a working volume of 0.002 m³ [33]. The main challenges in biomass production by *Y. lipolytica* at large scale might be to find the right level of oxygen, identify the optimal C/N molar ratio,

eliminate the formation of by-products such as citric acid, prevent extreme foam formation, which requires the use of a large amount of antifoam agents, and grow and produce biomass at higher temperatures, which has the potential to reduce the costs of the process by avoiding the use of cooling systems [32].

The aim of this study was to valorise WB into yeast biomass. For that purpose, WB was first pre-treated by enzymatic hydrolysis before being used as substrate to grow *Y. lipolytica* strain K57 in a batch bioreactor. To date, as far as we know, there are no reports on the production of yeast biomass from waste bread by *Y. lipolytica*.

## 2. Materials and Methods

### 2.1. Materials

#### 2.1.1. Microorganism

Wild-type *Y. lipolytica* strain K57 was used for bioreactor fermentations. It was obtained from the culture collection of the Department of Food Engineering, Faculty of Engineering, University of Ankara, Turkey. It was kept on yeast extract peptone dextrose agar (YPD) medium at 4 °C for a maximum duration of 2 weeks.

#### 2.1.2. Preparation of Waste Bread Powder

Waste bread (WB) was collected from local bakeries with mould-free and stale status. Before each hydrolysis, stale breads were manually cut into cubes of 2–4 cm in size. Obtained cubes were then dried in a drying oven at 50 °C for 24 h and ground in a kitchen-type blender. Waste bread powder (WBP) was stored in an airtight jar at room temperature until use [8].

#### 2.1.3. Enzymes and Antifoam

Termamyl SC DS alpha-amylase (thermostable $\alpha$-amylase from *Bacillus licheniformis* with declared activity of 300 U/mL), SAN Extra L (glucoamylase from *Aspergillus niger* with declared activity of 300 U/mL) and Neutrase 0.8 L protease (bacterial protease from *Bacillus amyloliquefaciens* with declared activity was 0.8 U/g) were used for waste bread hydrolysis [8]. One unit of enzyme activity was defined as the amount of enzyme releasing 1 µmol of reaction product per minute under the optimum conditions (U mL$^{-1}$). Antifoam Struktol J 673 (Schill+Seilacher "Struktol" GmbH, Germany) was used for bioreactor fermentations.

### 2.2. Hydrolysis of Waste Bread Powder

The optimization of waste bread powder hydrolysis was first performed in 250 mL shake flasks in a water bath under agitation. The total volume of WBP mixture used in the flask scale was 150 mL. Trials for small-scale hydrolysis were performed according to the method developed by Pietrzak and Kawa-Rygielska [8], using the same enzymes but with different enzyme loading quantities to find the best conversion rate of starch into glucose. For the large-scale hydrolysis, WBP hydrolysis was performed in a bioreactor (Bioflo 110 twin bioreactor, New Brunswick Scientific, USA) as illustrated in Figure 1. Briefly, 300 g of WBP was added to 1440 mL of distilled water and the pH of mixture was adjusted to 6.0 by the addition of 1 M of NaOH or 1 M of $H_2SO_4$. Reactor agitation and temperature were set at 500 rpm and 45 °C, respectively. The temperature was increased to 60 °C in 20 min and 0.6 mL of alpha-amylase solution (1 U/g WBP) was added, and the mixture was further heated up to 85 °C during 30 min and kept at this temperature for 60 min. After that, the temperature of the mixture was decreased to 55 °C (approx. in 20 min) and the pH was adjusted to 5.8 by adding 1 M $H_2SO_4$. In total, 3 mL of gluco-amylase solution (3 U/g WBP) and 0.263 mL of protease solution ($8.4 \times 10^{-4}$ U/g WBP) were added. After 90 min of incubation at 55 °C, the temperature was decreased to 20 °C (approx. in 30 min) and the pH was adjusted to 4.5 by the addition of 7.11 M $H_2SO_4$. The WBP hydrolysate was then weighed, and distilled water was added

to reach a final weight of 2 kg. Finally, it was clarified by centrifugation at 12,000 rpm for 30 min.

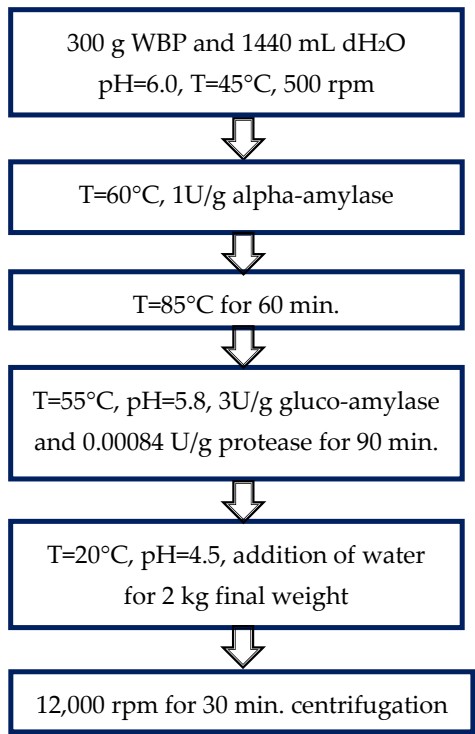

**Figure 1.** Hydrolysis of waste bread.

Lyophilisation

To increase the concentration of glucose, hydrolysate was lyophilised at −80 °C by using a freeze-dryer (ilShin FD8512, BX Ede, Netherlands). In total, 80% of the water was removed by this process.

### 2.3. Culture Conditions

The yeast inoculum, approximately $10^{10}$ cells, was prepared in a shake-flask with 50 mL medium consisting of 60 g/L glucose, 0.5 g/L ammonium sulphate, 0.5 g/L yeast extract, 7 g/L $KH_2PO_4$, 2.5 g/L $Na_2HPO_4$, 1.5 g/L $MgSO_4.7H_2O$, 0.15 g/L $CaCl_2$, 0.15 g/L $FeCl_3.6H_2O$, 0.02 g/L $ZnSO_4.7H_2O$, and 0.06 g/L $MnSO_4.H_2O$, and incubated in an orbital shaker at 185 rpm and a temperature of 28 °C for 48 h [19]. The batch bioreactor fermentation medium was similar to the seed-flask medium, containing waste bread hydrolysate (WBH) as the carbon source instead of pure glucose. WBH and salt solution [34] (same salt composition as in the shake-flask medium) were autoclaved separately at 121 °C for 15 min, and then aseptically combined to obtain medium containing 68, 101, and 142 g/L glucose. Bioreactor process parameters were set for agitation at 800 rpm, aeration at 1.0 vvm (volume of air per volume of medium), temperature at 28 °C and a pH range between 5 and 6 (addition of 5 M KOH periodically) during bioreactor fermentation. Cultures were carried out until all glucose was consumed by yeast. Temperature, pH (pH probe, Metler Toledo, Hong Kong) and $dO_2$ (dissolved oxygen sensor, Metler Toledo, Hong Kong) were monitored over time.

### 2.4. Analytical Methods

Biomass was determined as dry cell weight (DCW) and optical density ($OD_{600}$). Samples collected from culture medium were centrifuged at 5000 rpm for 15 min,

washed twice with distilled water and dried at 105 °C for 12 h until reaching constant weight for dry cell weight determination. $OD_{600}$ was measured at 600 nm in a UV/VIS spectrophotometer (Perkin Elmer Lambda 25, USA). Organic acids and glucose content in culture supernatant were determined by HPLC (Shimadzu brand LC-20AD model), equipped with a Bio-Rad HPX-87H (300 × 7.8 mm) column. Organic acids were detected using an ultraviolet detector (SPD-20A model), whereas sugars were detected using an RID (RID-10A model) detector. A total of 20 μL of supernatant was injected to the HPLC column and eluted at 50 °C at a flow rate of 0.5 mL/min using a 5 mM $H_2SO_4$ solution as a mobile phase. Free alpha amino nitrogen content (FAN) was determined according to EBC analysis method [35]. Quantity of total sugars in WB was determined according to Miller [36] and defined as glucose. Total raw protein content of waste bread powder (WBP) was determined according to the Kjeldahl method [37]. Total fat of WBP was determined by the Soxhlet method [38]. Moisture content of WBP was determined via the drying process in an oven (105 °C for 12 h) and ash content of WBP was examined via burning in a furnace at high temperatures (max. 650 °C). After the characterization of the chemical composition of WBP (Table 1), starch content was calculated considering that 95% of the total carbohydrate composition was composed of starch. Total lipid of collected biomass was determined according to Folch et al. [39]. Cellular lipids were extracted from biomass using 15 mL of a 2/1 (*v/v*) chloroform/methanol mixture. The lower moisture content in waste bread powder is the consequence of the drying process (at 50 °C in drying oven) of WB prior to grinding. Each analysis was performed in duplicate.

**Table 1.** Chemical composition of waste bread powder (N = 3).

| Composition | % |
|---|---|
| Ash | 1.30 ± 0.03 * |
| Crude Protein | 11.99 ± 0.18 * |
| Crude Fat | 0.71 ± 0.38 * |
| Total Carbohydrate | 82.01 ± 0.19 * |
| Moisture | 3.99 ± 0.04 |

* corresponds to dry matter.

*2.5. Statistical Analysis*

Obtained data were analysed using the statistical program SPSS (v.20) for Windows 8 (IBM, Armonk, NY, USA) with analysis of variance (ANOVA) and multiple range tests. Principle component analysis (PCA) was performed to simplify the interpretation of the results using the statistical program PAST (v.3.21) (University of Oslo, Oslo, Norway).

## 3. Results and Discussion

*3.1. Hydrolysis of Waste Bread*

The hydrolysis of waste bread was performed to convert starch into glucose and protein to FAN according to the method developed by Pietrzak and Kawa-Rygielska [8], who investigated the effect of using granular starch hydrolysing enzymes and raw material pre-treatments in the small-scale production of waste bread hydrolysate (WBH). However, in the current study, a reactor with a 2 L working volume was used to achieve high quantities of WBH by slightly modifying the same method. Within this optimisation, 2 kg of mash (including 150 g WBP/kg mash equal to 168 g WBP/litre mash) was obtained and used as a feedstock for *Y. lipolytica* strain K57. Primary hydrolysis tests in 250 mL beakers were performed to achieve maximum glucose content using various quantities of enzymes. Then, large-scale hydrolysis was applied in a bioreactor, and the optimum enzyme concentrations were found to be 1 U/g substrate for alpha-amylase, 3 U/g substrate for glucoamylase (loading) and $8.4 × 10^{-4}$ U/g substrate for protease.

The results for WBH are shown in Table 2. Two different trials were undertaken by adding the same quantity of alpha-amylase and protease but varying glucoamylase ac-

tivity (1 and 3 U/g substrate). A glucose content of 125.75 g/L with a conversion yield from starch of approximately 97.10% and a FAN content of 127.88 mg/L, with a yield from total Kjeldahl nitrogen (TKN) of 3.97%, were obtained when 3 U/g of glucoamylase loading was used during hydrolysis. Due to higher conversion yields, a glucoamylase loading of 3 U/g was selected for further hydrolysis.

**Table 2.** Conversion yield values of waste bread hydrolysis. A: glucoamylase loading of 1 U/g substrate, B: glucoamylase loading of 3 U/g substrate, and the other 2 enzyme loadings kept constant (N = 3).

| | Starch to Glucose CY (%), Glucose Yield (g G/ g Substrate) | TKN to FAN CY (%), FAN Yield (mg FAN/g Substrate) | Glucose (g/L) | FAN (mg/L) |
|---|---|---|---|---|
| **A** | 63.60%, 0.49 g/g | 3.39%, 0.64 mg/g | 82.37±3.30 | 108.73±8.39 |
| **B** | 97.10%, 0.75 g/g | 3.97%, 0.76 mg/g | 125.75±4.35 | 127.88±5.27 |

The abbreviations used in the table are as follows: CY: conversion yield, FAN: free alpha amino nitrogen, TKN: total Kjeldahl nitrogen.

The conversion yield from starch to glucose was very high, in contrast to the conversion yield from TKN to FAN, which was low as compared to previous studies. For instance, by using glucoamylase and protease produced from the fermentation of *Aspergillus awamori* on wheat milling by-products, starch to glucose conversion yields and TKN to FAN conversion yields higher than 90% (*w/w*) and 40% (*w/w*), respectively, were achieved through the batch hydrolysis of flour-rich waste (flour-rich waste concentrations up to 205 g/L) [40]. In a study on the hydrolysis of WB for succinic acid production, the conversion yields for total sugar and FAN were reported as 0.47 (g total sugar/g substrate) and 2.55 (mg FAN/g substrate), respectively [16]. In another study on the hydrolysis of WB by enzymes produced from the solid-state fermentation of *Aspergillus awamori* and *Aspergillus oryzae*, the highest starch conversions of 96.6% and glucose yield of 0.52 g glucose/g WB were reported [12]. By the application of a similar hydrolysis method, 49.8 g/L of glucose and 284 mg/L of FAN were produced [11]. In addition, comparable to present work, a maximum glucose yield of 86% was achieved by applying the same hydrolysis method (i.e., with the enzyme combinations) through a two-stage hydrolysis process of liquefaction and saccharification [41].

The results show that WB hydrolysis resulted in a higher glucose content, but a lower FAN conversion yield as compared to reports in previous studies. A high glucose concentration in feedstock and thus in culture medium can be an important factor for achieving a high quantity of biomass production during the cultivation of *Y. lipolytica*. Relatively high carbon and nitrogen levels might enhance the production of biomass by *Y. lipolytica* instead of the production of citric acid and/or single-cell oil. In addition, the FAN content of 127.88 ± 5.27 mg/L can promote biomass accumulation since biomass production also requires a nitrogen source. According to the literature, nitrogen concentrations above $10^{-3}$ mol/L (14 mg/L nitrogen) in the medium can increase biomass production, reduce lipid accumulation, and inhibit citric acid production [42].

### 3.2. Bioreactor Fermentation

Freeze-dried WBH was diluted in water to obtain a glucose content of 50, 100 and 150 g/L. This corresponds to a FAN content of approximately 65, 100 and 135 mg/L (Table 3). These three glucose concentrations were selected to compare the growth ability of *Y. lipolytica* in a bioreactor with previous studies performed on glucose-based media.

**Table 3.** Glucose and FAN concentrations of WBH used in bioreactor experiments.

| | Glucose (g/L) | FAN (mg/L) |
|---|---|---|
| 1 | 49.42 ± 1.98 | 65.23 ± 5.03 |

| | | |
|---|---|---|
| 2 | 96.82 ± 3.35 | 98.47 ± 4.06 |
| 3 | 146.00 ± 5.95 | 134.53 ± 5.47 |

FAN: free alpha amino nitrogen.

After the addition of the cell inoculum and other medium components to WBHs, initial glucose concentrations in culture were determined as 68.0, 101.2 and 142.5 g/L. Values of biomass, glucose, citric acid, and intracellular lipid accumulation were monitored over time for the three experimental conditions tested (Figures 2–4). One of the differences between the bioreactor trials on pure-glucose-based and WBH-based media [19] was that the cell growth time was shorter in the WBH-based culture. Indeed, glucose was depleted in the medium after 36, 47 and again 47 h from the starting concentrations of 68, 101 and 142 g/L, respectively.

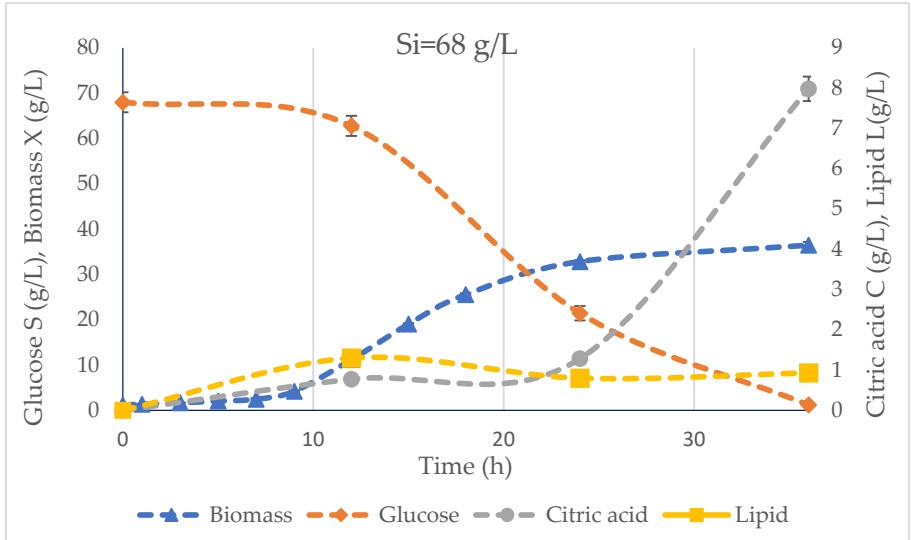

**Figure 2.** Kinetics of cell growth, glucose consumption, production of CA and lipid accumulation for *Y. lipolytica* strain K57 during growth on waste bread hydrolysate media at an initial glucose concentration of 68 g/L under a limited nitrogen concentration at an agitation of 800 rpm and aeration of 0.5 vvm. Incubation time of 36 h.

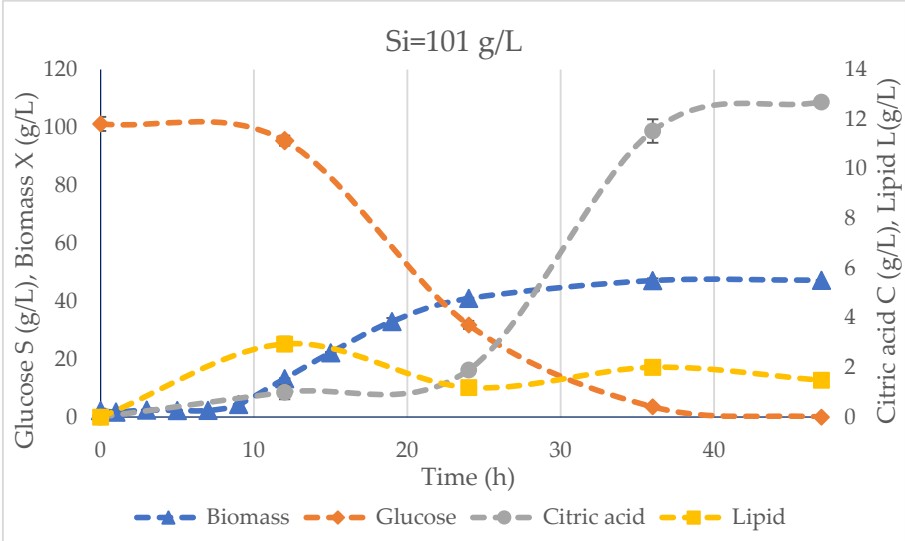

**Figure 3.** Kinetics of cell growth, glucose consumption, production of CA and lipid accumulation for *Y. lipolytica* strain K57 during growth on waste bread hydrolysate media at an initial glucose

concentration of 101 g/L under a limited nitrogen concentration at an agitation of 800 rpm and aeration of 0.5 vvm. Incubation time of 47 h.

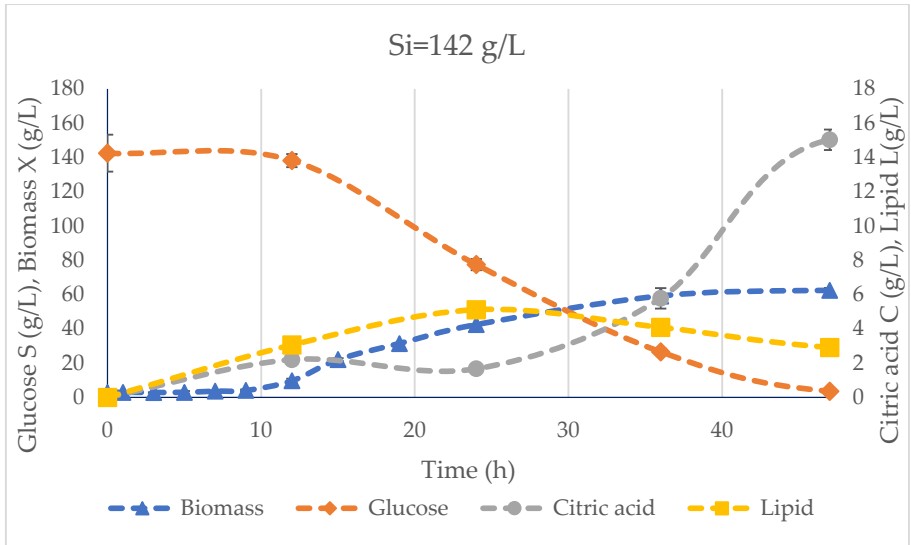

**Figure 4.** Kinetics of cell growth, glucose consumption, production of CA and lipid accumulation for *Y. lipolytica* strain K57 during growth on waste bread hydrolysate media at an initial glucose concentration of 142 g/L under a limited nitrogen concentration at an agitation of 800 rpm and aeration of 0.5 vvm. Incubation time of 47 h.

In addition, CA production was lower while biomass content and lipid production were higher in WBH fermentations compared to glucose-based media [19]. CA concentrations ranged between 7.99 ± 0.30 g/L and 15.03 ± 0.59 g/L, the biomass content varied between 36.44 ± 0.75 g/L and 62.48 ± 1.43 g/L, and the maximum lipid content ranged between 1.30 ± 0.03 g/L and 5.12 ± 0.11 g/L during bioreactor runs with three different glucose concentrations of WBH (68.0, 101.2 and 142.5 g/L, respectively) Table 4.

**Table 4.** Kinetic parameters of *Y. lipolytica* strain K57 on three different glucose concentrations of a waste bread hydrolysate in a bioreactor, including the maximum achieved lipid and CA contents. The kinetic parameters presented in Table 4 were calculated according to the equations referred to in Dlangamandla et al., 2019 [43].

| Parameters | Si = 68 g/L Value | Si = 101 g/L Value | Si = 142 g/L Value | Unit |
|---|---|---|---|---|
| Glucose consumption (S) | 66.78 ± 0.03 | 101.23 ± 0.00 | 138.95 ± 0.00 | g/L |
| Biomass (X) | 36.44 ± 0.75 | 47.23 ± 0.83 | 62.48 ± 1.43 | g/L |
| Acetic acid production | 1.35 ± 0.06 | 1.49 ± 0.09 | 3.30 ± 0.29 | g/L |
| CA production (C) | 7.99 ± 0.30 | 12.69 ± 0.08 | 15.03 ± 0.59 | g/L |
| Yield biomass ($Y_{X/S}$) | 0.55 ± 0.01 | 0.47 ± 0.01 | 0.45 ± 0.01 | g/g |
| Yield citric acid ($Y_{C/S}$) | 0.12 | 0.13 | 0.11 | g/g |
| Consumption rate ($R_S$) | 1.29 | 3.00 | 2.46 | g/L/h |
| CA production rate ($Q_C$) | 0.19 | 0.34 | 0.30 | g/L/h |
| Max. consumption rate ($R_{Smax}$) | 3.44 | 5.30 | 5.06 | g/L/h |
| Max. production rate ($Q_{Cmax}$) | 0.56 | 0.80 | 0.84 | g/L/h |
| Max. growth rate ($q_{max}$) | 2.47 | 2.20 | 2.49 | g/L/h |
| Specific growth rate ($\mu$) | 0.30 | 0.30 | 0.37 | 1/h |
| Doubling time ($t_d$) | 2.28 | 2.30 | 1.87 | h |
| Max. lipid production ($L_{max}$) | 1.30 ± 0.03 | 2.95 ± 0.29 | 5.12 ± 0.11 | g/L |
| Max. lipid yield ($Y_{(L/S)max}$) | 0.12 | 0.22 | 0.32 | g/g |
| C/N molar ratio | 29 | 40 | 49 | mol/mol |

Si: initial glucose concentration.

For the tested glucose concentrations of 68 and 101 g/L, strain K57 on WBH-based media slightly grew during the first 24 h, then started to grow exponentially up to 36 h. For the glucose concentration of 142 g/L, the cell growth was delayed, possibly due to the high glucose concentration (osmotic stress effect). High glucose concentration in the growth medium has a negative effect on cell growth due to osmotic stress and the inhibition of specific enzymes in cell metabolism. DO concentration is one of the main factors affecting biomass production; as DO concentration in the medium increases, biomass production also increases. At the 18th h of culture, DO values were the lowest, with values of 30%, 11% and 1.78% for the initial glucose concentrations of 68, 101 and 142 g/L, respectively, before remaining at constant values that ranged between 80 and 85%. So, this means that the DO value was adequate for CA production by *Y. lipolytica* in these conditions. However, it may be better to increase oxygen transfer rate conditions during the time interval from the 12th to the 18th h of culture, since at this time DO values were lower than 20% for higher initial glucose concentrations (101 g/L and 142 g/L), which were reported as the critical DO % for biomass production in previous studies [32].

Very high biomass concentrations of 36.4 g/L, 47.2 g/L and 62.5 g/L were obtained from initial glucose concentrations of 68 g/L, 101 g/L and 142 g/L, respectively. Previous studies in bioreactors with glucose-based media reported that the highest biomass content was about 5.0 g/L [19]. This difference in the accumulation of biomass could be explained by the positive effect of the nitrogen content of WBH on the growth of *Y. lipolytica* strain K57 [16,44]. WBH with a high FAN concentration, in addition to ammonium sulphate and yeast extract (Sigma-Aldrich, Missouri, USA), led to an increase in the growth performance and thus biomass formation of *Y. lipolytica* strain K57. Moreover, WBH also had a high crude protein content at approximately 12% dry matter (Table 1). WB is rich in carbon and nitrogen content, which makes it a very nitrogenous substrate for the fermentation process. A previous study on lipid production from flour-rich waste by *Lipomyces starkeyi* showed that a high value of initial glucose concentration (110 g/L), and a FAN content of 140 mg/L in shake-flask fermentation led to the accumulation of high biomass content (30.5 g/L) [40]. In addition, in the same study, *L. starkeyi* biomass reached 113 g/L in a fed-batch bioreactor culture from a medium containing 90 g/L of glucose and 180 mg/L of FAN. Similarly, in our study, 47.15 g/L of *Y. lipolytica* biomass was obtained after 36 h of batch bioreactor culture in medium containing 101.23 g/L of glucose and about 100 mg/L of FAN. In another study on single-cell oil (SCO) production in bioreactor batch culture by *R. toruloides,* 30 g/L of biomass was obtained when flour-rich waste with 100 g/L glucose and 200 mg/L FAN was used as a medium [45]. According to these results, it can be concluded that the biomass values obtained in the present study are in agreement with the reported values in the literature.

Glucose and FAN concentrations in WBH affected the biomass production by *Y. lipolytica*. A high FAN content favours biomass formation. The highest biomass value of 62.48 g/L was obtained for initial glucose and FAN concentrations of 142 g/L and 130 mg/L. Moreover, this high initial glucose concentration did not inhibit the growth of *Y. lipolytica* strain K57. Increased biomass content is favoured for lipid production, and as dry cell weight increases, the produced lipid content also increases. However, this condition is not preferable for CA productivity as the carbon source in the medium can be further used for cell growth and lipid production instead of CA synthesis.

CA production by *Y. lipolytica* strain K57 in WBH medium was lower as compared to bioreactor trials of glucose-based media [19]. The highest CA amount of 15.03 g/L was produced from WBH with 142 g/L of glucose at the end of the 47th h of batch reactor fermentation. However, in the same type of fermentation with glucose-based media, the highest CA of 72.12 g/L was produced from an initial glucose concentration of 101 g/L [19]. The main reason for this result can be explained by extreme biomass formation, which led to the accumulation of low concentrations of CA. Various industrial by-products have been used as a substrate for CA production by *Y. lipolytica* strains; of these, the most commonly used substrate was glycerol. Agricultural waste such as olive

mill wastewater, animal and vegetable fats, and waste cooking oil have been used as carbon sources for CA production by *Y. lipolytica*, but these by-products contained a lower amount of nitrogen content. WB consists of a high nitrogen content with a FAN value of more than 100 mg/L and a protein content of around 10% dry matter. In a study performed with whey as substrate for CA production by *Y. lipolytica*, a lower amount of CA was produced due to a high content of ammonium chloride, yeast extract and crude protein in the whey medium, which may have a negative effect on CA production [46]. Similarly, in another study of CA production from extract of Jerusalem artichoke tubers by *Y. lipolytica*, a CA titre of 68.3 g/L was obtained in a 10-litre batch bioreactor culture within 336 h. The addition of ammonium sulphate to the Jerusalem artichoke tuber extract resulted in a decrease in CA production and an increase in biomass [47]. In our study, higher nitrogen compounds (FAN, ammonium sulphate and yeast extract) were also present in the fermentation medium with WBH, which could lead to reduced CA production.

Figures 2–5 show the intracellular lipid contents and lipid yield in cell biomass for *Y. lipolytica* grown on WBH-based media. The highest lipid contents were observed between the 12th and 24th h of fermentation and decreased afterwards. According to the lipid content profile, during the first 24 h, cell growth was low and lipid accumulation started in cells. Then, the carbon source was used for cell growth instead of lipid production, as lipid production decreased while exponential cell growth was observed after the 24th h of fermentation (Figures 2–4). The highest value of biomass/lipid conversion yield (0.32 g lipid/g biomass) was observed at 12 h of cell growth on medium containing an initial glucose concentration of 142 g/L. Moreover, after 24 h of fermentation, the highest lipid titre (of 5.12 g/L) was obtained from WBH for an initial glucose concentration of 142 g/L. From these results, it can be concluded that the higher initial glucose concentrations of WBH with a higher C/N molar ratio of 49 promoted increased lipid content and lipid yield during the initial phase of culture (i.e., the first 24h). However, at the later stage, the lipid in the cell biomass did not increase, as observed in cell growth. In contrast to these results, the lipid yields of oleaginous yeast reported in other studies are generally significantly lower during the initial hours of cultivation with a lower C/N molar ratio. At the later phase of cultivation, mainly due to nitrogen exhaustion, the lipid in dry cell biomass increases at a high C/N molar ratio. However, this also depends on oleaginous yeast strain, since a previous study on the production of lipid by *Y. lipolytica* strain K57 cultivated on glucose in batch cultures at a C/N molar ratio of 52 showed that the maximum lipid yield obtained was 0.09 g lipid/g biomass. A previous experimental study in *S. cerevisae* biomass production from sugarcane molasses during a glucose-limited feeding strategy showed that the overall biomass yield on substrate of 0.33 (g/g) and a specific growth rate of 0.16 (h$^{-1}$) were obtained [48]. By contrast, in the current study, the biomass yield ranged between 0.45 and 0.55 (g/g) and the specific growth ranged between 0.30 and 0.37 h$^{-1}$, which was achieved using different initial glucose concentrations of WBH. In conclusion, compared to baker's yeast, a higher yield of biomass and a higher growth rate were obtained by *Y. lipolytica* when WBH was used as feedstock.

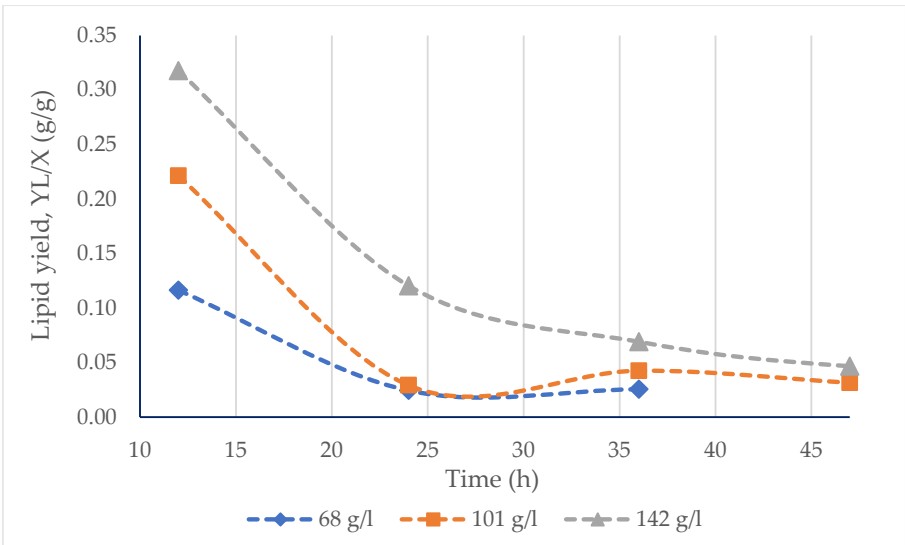

**Figure 5.** Lipid yield (g lipid/g biomass) results during batch bioreactor fermentation for the condition with an initial glucose concentration of 68 g/L (blue line), 101 g/L (orange line) and 142 g/L (grey line).

It has also been observed in other studies that higher lipid concentrations were generated by increasing the initial glucose concentrations. In previous studies, hydrolysate of wheat straw was used for lipid production by *Y. lipolytica* [49], and less than a 1.0 g/L lipid titre and 4.5% (*w/w*) lipid yield with a biomass formation of 7.2 g/L were produced by 6-day shake-flask fermentation. The initial glucose concentration of the wheat straw hydrolysate in that study was very low, with a value of approximately 3.5 g/L, and some other monosaccharides, such as xylose, arabinose, and galactose, were present in the fermentation medium. In another work, Tsigie et al. [50] obtained a lipid content of 6.7 g/L and yield of 0.58 g/g DCW from a medium based on detoxified sugarcane bagasse hydrolysate and peptone (initial total sugar available concentration of 20 g/L and peptone of 5 g/L) by *Y. lipolytica* Po1g via shake-flask fermentation. Moreover, the same researchers also studied lipid production from a detoxified rice bran hydrolysate by *Y. lipolytica* Po1g [51] and produced a DCW of 10.75 g/L and lipid content of 48.02%. In the same study, the effect of the nitrogen content and initial glucose concentration of a detoxified rice bran hydrolysate on lipid productivity was also examined, and the highest lipid productivity was found to without the addition of any nitrogen source to a medium containing an initial glucose concentration of 30 g/L instead of 20 or 40 g/L. With a high content of nitrogen use in the medium, biomass formation and glucose consumption increased, and a level of more than 30 g/L of initial glucose present in the medium inhibited the growth of *Y. lipolytica* Po1g. In our study, a high nitrogen content of the medium also stimulated biomass formation and glucose consumption. Ratledge and Wynn [52] explained that as the nitrogen source is exhausted in the medium, excess carbon is converted to lipid production by oleaginous microorganisms. Therefore, the key factor for WBH hydrolysis may have been high nitrogen content, which can promote high biomass formation and fast glucose consumption, causing less lipid and CA production. According to the results obtained in previous studies, it can be concluded that a maximum lipid yield value of 0.32 g/g DCW with 5.12 g/L lipid content (Table 4) was found at the WBH initial glucose concentration of 142 g/L, which is comparable to values reported in the literature. Very high biomass formation and fast glucose consumption were observed.

*3.3. Principle Component Analysis (PCA) Results of Kinetics Parameters for Y. lipolytica K57 on a Waste Bread Hydrolysate Media in a Reactor Fermentation*

To determine the effect of different initial glucose concentrations on fermentation kinetics for *Y. lipolytica* K57 on waste bread hydrolysate media, PCA analysis was performed. The total variance explained by two principal components was 92.37%, with the first component (F1) comprising 65.80% and the second component (F2) 26.57%. Figure 6 illustrates the bi-plot of PCA, composed of two distinct groups with regard to F1. The first one was generated by samples 3a-12h (Si of 142 g/L, at 12h and at max. lipid yield), 2a-12h (Si of 101 g/L at 12 h and max. lipid yield) and 1a-12h (Si of 68 g/L at 12 h and max. lipid yield), characterized by CA yield, lipid, lipid yield, biomass yield and acetate. The second group was formed by samples 3b-47h (Si of 142 g/L, at 47 h and at max. CA content), 2b-47h (Si of 101 g/L, at 47 h and at max. CA content) and 1b-36h (Si of 68 g/L, at 36 h and at max. CA content), explained by CA, consumed glucose and biomass. As can be seen from Figure 6, sample 3a-12h is obviously better described by lipid yield, CA yield, and biomass yield, while sample 3b-47h is determined mostly by CA, biomass and consumed glucose. Consequently, it can be concluded that the highest initial glucose concentration of 142 g/L had a better effect on biomass production.

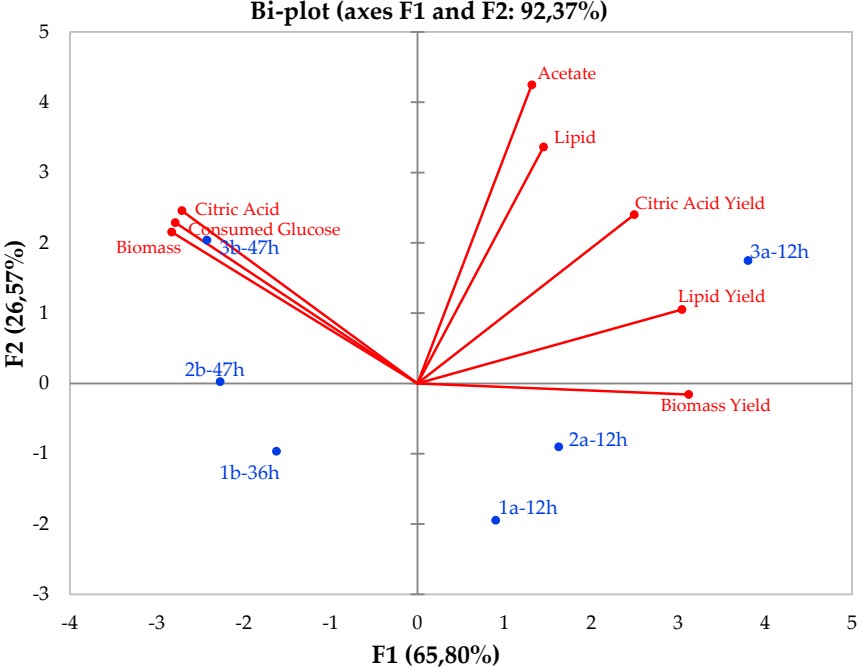

**Figure 6.** Bi−plot of PCA regarding fermentation kinetics for *Y. lipolytica* K57 on WBH media in a reactor fermentation (1: Si = 68 g/L, 2: Si = 101 g/L, 3: Si = 142 g/L, a: at the max. lipid, b:at the max. CA, h: hour).

## 4. Conclusions

This study showed that with the effective enzymatic hydrolysis method in a bioreactor, a high amount (around 2 kg) of waste bread hydrolysate with a very high glucose concentration (125.75 g/L) can be generated in a very short period of time (approx. 5 h). Waste bread hydrolysate can be used as alternative carbon source for the growth and biomass production of *Y. lipolytica* strain K57. A relatively high amount of yeast biomass (62 g/L) with the biomass yield of 0.45% (g DCW/g glucose) was achieved, which can be considered as potential additive for animal and human dietary products. Thus, waste bread hydrolysate can be successfully used to produce yeast biomass with *Y. lipolytica* strain K57.

**Author Contributions:** Conceptualization, H.E., S.P., P.F. and E.C.; methodology, H.E. and E.C.; software, E.C.; validation, E.C., H.E. and B.A.; formal analysis, E.C.; investigation, E.C.; resources, H.E.; data curation, E.C. and B.A.; writing—original draft preparation, E.C.; writing—review and editing, B.A., H.E., S.P. and P.F.; visualization, H.E. and E.C.; supervision, H.E. and S.P.; project administration, H.E.; funding acquisition, H.E. and E.C. All authors have read and agreed to the published version of the manuscript.

**Funding:** This research was funded by the Scientific Research Projects Unit (BAP) of Cukurova University (project no: FDK-2014-3159), Adana, Turkey.

**Institutional Review Board Statement:** Not applicable.

**Informed Consent Statement:** Not applicable.

**Data Availability Statement:** The essential data supporting the reported results are contained in this study. Additional data related to this paper are available on request from the corresponding author.

**Conflicts of Interest:** The authors declare no conflicts of interest.

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
