# Peer review of "Valorisation of Waste Bread for the Production of Yeast Biomass by Yarrowia lipolytica Bioreactor Fermentation"

_fermentation, doi:10.3390/fermentation9070687_

Round 1

Reviewer 1 Report

a)-The authors did not specify how they defined a unit of enzyme activity in the text.

b)- The other point is whether the bread they collect from local bakeries has a cost and continuous supply. If this component is recommended for use in industrial yeast production, it should be provided from an inexpensive and reliable source. For example, the advantages and disadvantages of the waste bread (e.g., cost, yield, suitability for the applied process etc.) can be compared with the materials currently used for the same purpose (e.g., sugarcane molasses).

c)- Authors can write a future perspective section. Here, it can be suggested to investigate how WB hydrolysate (WBH) will work under higher-scale fermentation conditions. Because it is well known that the parameters determined for small-scale fermentation (i.e., 2 L) sometimes do not work in pilot and large-scale applications. For example, what size fermenter is Yarrowia lipolytica grown in industrial applications?

The study is generally well explained, but it is not a high-impact study.

Line 47)- what nitrous means?

Section 2.1.3. How the authors determined the optimal ratios of 3 enzymes mixture for hydrolysis products from bread because the product of one enzyme inhibits the other enzyme (amylase vs glucoamylase). Protease will digest amylase and glucoamylase. Reference 5 does not say anything. Was a productivity analysis of all 3 enzymes carried out? Include a short description about the issue in view of the following review on enzymatic productivity analysis. Evaluating Enzymatic Productivity—The Missing Link to Enzyme Utility. Int. J. Mol. Sci. 2022, 23(13), 6908; https://doi.org/10.3390/ijms23136908

Figure 1 scheme has some steps that include high temperature such as 85 deg C. This is high enough to hydrolyze glycosidic bonds in starch in the absence of enzymes. Explain why non-enzymatic hydrolysis was not followed as a control. That is another reason that the first step (Enzymatic hydrolysis) should have been optimized using productivity curves. Explain this issue in the revised manuscript along with the above point re Section 2.1.3.

 Citric acid is a useful industrial product but its importance was not considered beneficial though it is produced in high amounts in this study. Explain.

Is OK. needs minor corrections.

Reviewer 2 Report

Introduction

1.     Lines 42-43: “Such a nutrient-rich food waste, WB is harmful to the environment…” The reviewer disagrees with this statement. WB can not be considered as harmful to the environment as some chemical substances released from industrial production processes such as chemical, petrochemical, textile, etc. Please revise the sentence.

2.     Line 69: “ The aim of this study was to valorise WB into yeast biomass”. Did you mean “to valorise conversion of WB into yeast biomass”? The authors should consider rewriting the sentence.

Materials and methods

3.     Section 2.2.: Please describe enzyme hydrolysis optimisation at a small scale (in 250 mL shake flasks).

4.     Line  125: “WBH and salt solution were autoclaved separately….”  Please add the composition of the salt solution.

5.     Dissolved oxygen is critical for cell biomass growth and metabolite accumulation. Since the dissolved oxygen was monitored, the authors should write its value during the cultivation.

6.     The authors should add the equations for calculating specific kinetic parameters presented in Table 4 or add the reference.

7.     Line  155: Please write the composition of chloroform/methanol solution (add units, e.g. vol/vol or w/w).

8.     Table 1. The reviewer assumes that ash, crude protein, crude fat and total carbohydrate content were calculated on wett biomass since the remaining 3.99 % of biomass equals 3.99 %, e.i. content of moisture in biomass. Please comment on this and revise the table accordingly.

9.     Results and discussion

10.  Line 278, instead of “osmatic” should be “osmotic”.

11.  Lines 277-278: High glucose concentration in the growth medium has a negative effect on cell growth due to osmotic stress and inhibition of specific enzymes in cell metabolism. Please revise the text carefully.

12.  Line 165: The section's name is Results and Discussion. Please correct

13.  Section 3.2.

The authors lyophilised the WBH and further dissolved it in water to prepare the cultivation medium. Is it feasible to use an energy-intensive method to concentrate the WBH since the value of the product (cell biomass) is relatively low? The concentration of glucose and FAN in the growth medium with the highest glc and FAN concentration is slightly higher than concentrations of glc and FAN in WB (Table 2). Please comment on this.

14.  Figures 2-4 Legends “ Kinetics of biomass, citric acid,…”  Please rewrite the legend, e.g. “The presented values in the figure are concentrations of cell biomass, citric acid and glucose….” or “ Kinetics of cell growth, glucose consumption, production of CA and lipid accumulation…”

15.  Lines 332-350

First, the reviewer would suggest presenting lipid content in cell biomass in addition to lipid titer or concentration. Also, it would be useful to determine nitrogen content (FAN and Kjeldal nitrogen) during the cultivation and present it in Figures 2-4. This would help explain carbon flow in cell metabolism, the yield of citric acid and lipid content. According to the presented results in Figures 2-4. lipid content in cell biomass was generally below 15 %, which agrees with the literature data. The rather high value of lipid content (32 %, w/w) was observed only at the 12th hour of cultivation (Figure 4), while the lipid content at other sampling times was significantly lower. Since the fermentations were conducted once (missing mean values and standard deviations at cultivation curves in Figures 2-4), one could assume that this high value of lipid content was a consequence of analytic or systematic error. Furthermore, lipid content in cell biomass of lipid accumulating yeast presented in literature is significantly lower during the initial hours of cultivation in rich fermentation media (lower C/N) such as WBH. In the later phase of cultivation, mainly due to nitrogen exhaustion, the lipid content increases in the presence of a carbon source (high C/N). Please comment on the lipid content in the cell biomass.

Reviewer 3 Report

I am very grateful you for the invitation to review manuscript fermentation-2478000 by Carsanba and coauthors "Valorisation of Waste Bread for the Production of Yeast Biomass by Yarrowia lipolytica Bioreactor Fermentation”. The aim of this study was to test waste bread as a feedstock for biomass production by Yarrowia lipolytica. The work is interesting but needs several adjustments to increase the quality of the material.

Comments:

- Abstract, Line 14: Please specify the information regarding the increase in waste.

- Abstract: Please make it clear that the solution would be a waste reduction, and only then other uses.

- Lines 14-18: Up to this point in the work its application is not clear. The information is generic.

- Line 19: And what is the purpose of producing Yarrowia biomass?

- Line 25-26: The conclusion presented is not as clear as presented by the authors. The objective of the work needs to be revised and deepened.

- Lines 27-28: Change the repeated keywords by different words from the title.

- Line 37: The return rate of products to manufacturers should be highlighted.

- Introduction: It should be improved, highlighting the primary option of reducing waste, so that it can be applied to other processes.

- Line 29: The purpose of yeast production is unclear. Do the authors only want biomass production?

- Please remove the "." abbreviation of minutes throughout the text.

-  Lines 106-107 and throughout the text: Standardize the use of minutes and min.

- Line 111: Using a drying process that is so energy dependent is contradictory to environmental issues. Several more efficient processes can be applied.

- Results: The presentation of the results is superficial and there is no discussion about the biochemical mechanisms, mainly related to the material conversions.

- Line 394: Where is the "optimization"?

- Results: The authors do not present comparative information to produce a widely used yeast.

- Discussion: Where is the discussion?

- The conclusion, as well as the other items, do not justify the real objective (Production of biomass by biomass?).

Round 2

Reviewer 1 Report

The authors have taken into account all the comments and suggestions in the revised manuscript.

Can be checked during editing/formatting by the journal.

Reviewer 3 Report

The authors significantly corrected the manuscript according to the reviewer's suggestions.